# Is yearly interferon gamma release assay latent tuberculosis infection screening warranted among patients with rheumatological diseases on disease-modifying drugs in non-endemic settings?

**Carlo Foppiano Palacios**[ID][1]*, **Vaidehi Chowdhary**[2], **Ritche Hao**[1], **Abhijeet Danve**[2], **Maricar Malinis**[ID][1,3]*

**1** Section of Infectious Diseases, Department of Internal Medicine, Yale University School of Medicine, New Haven, Connecticut, United States of America, **2** Section of Rheumatology, Allergy and Immunology, Department of Internal Medicine, Yale University School of Medicine, New Haven, Connecticut, United States of America, **3** Department of Surgery (Transplant), Yale University School of Medicine, New Haven, Connecticut, United States of America

* foppianopalacios-car@cooperhealth.edu (CFP); Maricar.malinis@yale.edu (MM)

## Abstract

### Objective

Patients living with rheumatologic diseases on disease-modifying antirheumatic drugs (DMARD) are at an increased risk of developing tuberculosis (TB). Current guidelines recommend screening for latent tuberculosis infection (LTBI) before initiating DMARD. However, data is lacking on the value of yearly screening for LTBI.

### Methods

A retrospective chart review was conducted on adult patients ($\geq$ 18 years) with rheumatologic disease on DMARD followed longitudinally in the outpatient rheumatology clinics between 2017–2021. Collected data included patient demographics, rheumatologic diagnosis, medications, TB-related risk factors, interferon gamma release assay (IGRA) results, LTBI diagnosis and treatment. Descriptive statistics were performed.

### Results

Among 339 patients, 81 (23.9%) were male, 259 (76.4%) were white, and 93 (27.5%) were Latinx. Inflammatory arthritis (84.1%) was the most common rheumatic diagnosis. Common DMARD were JAK inhibitors (19.2%), TNF-alpha inhibitors (18.9%), and IL-17 A inhibitors (18.0%). Only 2 patients at baseline had positive IGRA, and both had a history of treated LTBI. Positive IGRA tests were recorded in 1 (0.7%), 3 (1.8%), 3 (1.3%), and 3 (1.1%) in the years 2018, 2019, 2020, and 2021, respectively. Four patients converted from negative to positive during serial yearly IGRA testing. After reviewing the IGRA test and TB risk factors, only one patient was considered newly diagnosed with LTBI, requiring 4 months of rifampin.

**Data Availability Statement:** All relevant data are within the manuscript and its Supporting Information files.

**Funding:** The authors received no specific funding for this work.

**Competing interests:** The authors have declared that no competing interests exist.

## Conclusion

In a non-endemic area, serial IGRA testing of low-risk patients on DMARD yielded very low rate of newly diagnosed LTBI. A targeted LTBI screening based on TB-related risk factors should be performed prior to IGRA testing rather than universal yearly screening in a non-endemic setting.

## Introduction

Patients living with rheumatologic diseases on disease-modifying antirheumatic drugs (DMARD) are at an increased risk of developing tuberculosis (TB), particularly with TNF-alpha inhibitors [1]. TNF-alpha, a target for some DMARD, is involved in granuloma formation and encases TB, thus protecting the host; hence, its inhibition increases the risk of TB reactivation [2]. Current data demonstrates that individuals with rheumatic diseases on TNF-alpha inhibitors in contrast with those not on DMARD have four times and ten times higher risk for LTBI reactivation in non-endemic and endemic settings, respectively [3].

Currently, the American College of Rheumatology (ACR) guidelines recommend that LTBI screening should be performed prior to DMARD initiation [4]. Screening for LTBI is performed with either tuberculin skin test (TST) or interferon-γ release assay (IGRA) [4]. Screening practices of patients on DMARD vary across centers depending on TB endemicity. Centers located in areas with high incidence of TB are recommended to perform LTBI screening annually [5]. However, there is lack of guidance on the value of subsequent IGRA-based screening after initial negative screening of patients living in TB non-endemic settings. Our center, which is in a non-endemic area for TB, performs yearly LTBI IGRA-based screening of patients with rheumatic disease on DMARD. We sought to evaluate the clinical utility of annual IGRA-based LTBI screening among this cohort.

## Materials and methods

### Data collection

We conducted a retrospective review of electronic medical record of patients age ≥ 18 years followed longitudinally in rheumatology clinic of the Yale-New Haven Hospital (New Haven, CT, USA) from 2017 to 2021. Patients who were prescribed biologics or targeted synthetic disease-modifying antirheumatic drugs (DMARD) and who had at least two IGRA assays done during study period were included. Disease-modifying antirheumatic drugs (DMARD) were grouped by mechanism of action (**S1 Table**). Data collected from the electronic medical record (EMR) (EPIC™, Wisconsin, USA) included the following: demographics, rheumatological diagnoses, medications, risk factors for TB, IGRA results, LTBI diagnosis, and, if applicable, LTBI treatment. Rheumatic diseases were grouped as inflammatory arthritis, connective tissue disease, vasculitis, idiopathic inflammatory myositis, polymyalgia rheumatica, or miscellaneous diseases (**S2 Table**). Most cases of TB reactivation in the US are more likely to occur in patients born outside the US. To identify these patients, we used primary spoken language as a surrogate marker in the EMR. Patients were classified as being born or having traveled to TB-endemic countries if the country had a TB incidence >10 per 100,000 per the World Health Organization classification [6]. IGRA assays utilized in our institution were QuantiFERON-TB Gold and QuantiFERON-TB Gold Plus [Qiagen, Netherlands] before and after July 17, 2018,

respectively. This study was approved by the Yale University Institutional Review Board and they waived the need for informed consent.

### Outcomes

The primary outcome was the proportion of patients with conversion from a negative to positive IGRA and a diagnosis of LTBI. The secondary outcome included identifying the risk factors for TB among patients with a negative to positive IGRA conversion.

### Statistical analyses

Descriptive statistics were performed. We conducted all data analysis and created figures with Microsoft Excel and R version 4.0.2.

## Results

### Patient demographics

While patient with rheumatic diseases on DMARDs were supposed to undergo yearly IGRA testing, data demonstrated 339 patients underwent IGRA testing at least twice during the study period including 51.9% (N = 176) with at least 3 IGRA tests, 18.6% (N = 63) with at least 4 IGRA tests, and 5.3% (N = 18) IGRA testing every year during the study period. Thus, only 339 patients were included in the analysis, 81 (23.9%) were male, 259 (76.4%) identified as white, and 93 (24.5%) were Latinx. The mean age of patients at the time of rheumatology visit in 2021 was 54.5 ±13.4 years (Table 1).

### Rheumatological history

Inflammatory arthritis (N = 286, 84.1%) was the most common rheumatic diagnosis (Table 1). The most common DMARD were JAK inhibitors (N = 65, 19.2%), TNF-alpha inhibitors (N = 64, 18.9%), and IL-17 A inhibitors (N = 61, 18.0%).

### Tuberculosis history and IGRA results

Fifteen patients (4.4%) had a documented history of LTBI in their EMR. At the start of the study period, only two tested positive for IGRA and had documented prior LTBI treatment. There were 1 (0.7%), 3 (1.8%), 3 (1.3%), and 3 (1.1%) positive IGRA tests in 2018, 2019, 2020, and 2021 respectively (S3 Table and S1 Fig). One patient in 2018 converted from negative to positive IGRA. Of the three patients in 2019 with positive IGRAs, two had negative to positive IGRA conversion and one had known previously treated LTBI. Of the three patients with positive IGRAs in 2020, two had prior treated with LTBI, and one had repeat negative IGRA and was adjudicated as not having LTBI. In 2021, one patient with positive IGRA had previously been treated with LTBI, one patient had a new negative to positive conversion, and the other had a previous positive IGRA in 2019.

The primary language of most patients was English (N = 326/339, 96.2%). Using having a primary language as a surrogate for foreign-born, only 13 patients (3.8%) spoke a primary language other than English. The most identified TB predisposing factors from the patient's problem list included diabetes (N = 49, 14.5%), tobacco use (N = 14, 4.1%), and heavy alcohol use (N = 4, 1.2%). Of the 18 patients who had documented completed IGRA testing every year during the study period, 4 (22.2%) had new travel to a TB-endemic setting.

**Table 1. Demographics among all patients (N = 339).**

|  | N | % |
|---|---|---|
| Age (years) | | |
| 20 and younger | 1 | 0.3 |
| 21 to 30 | 12 | 3.5 |
| 31 to 40 | 42 | 12.4 |
| 41 to 50 | 69 | 20.4 |
| 51 to 60 | 105 | 31.0 |
| 61 to 70 | 74 | 21.8 |
| 71 and older | 36 | 10.6 |
| Sex | | |
| Female | 258 | 76.1 |
| Male | 81 | 23.9 |
| Race | | |
| American Indian or Alaska Native | 1 | 0.3 |
| Asian | 11 | 3.2 |
| Black or African American | 32 | 9.4 |
| Other | 47 | 13.9 |
| White or Caucasian | 245 | 72.2 |
| Latino ethnicity | 57 | 16.8 |
| Primary language | | |
| English | 326 | 96.2 |
| Non-English | 13 | 3.8 |
| Rheumatic disease[a] | | |
| Inflammatory arthritis | 285 | 84.1 |
| Connective tissue disease | 58 | 17.1 |
| Vasculitis | 17 | 5.0 |
| Idiopathic inflammatory myositis | 7 | 2.1 |
| Polymyalgia rheumatica | 3 | 0.9 |
| Miscellaneous diseases | 18 | 5.3 |
| Disease Modifying anti-Rheumatic Drug (DMARD)[b] | | |
| TNF-alpha inhibitors | 64 | 18.9 |
| IL-6 and IL6-R blockers | 28 | 8.3 |
| T-cell co-stimulation inhibitor | 28 | 8.3 |
| B-cell depleting agents or Blys/BAFF inhibitor | 55 | 16.2 |
| IL-17 A inhibitors | 61 | 18.0 |
| IL-12/23 axis blockade | 43 | 12.7 |
| Cyclophosphamide | 3 | 0.9 |
| JAK inhibitor | 65 | 19.2 |
| Vedolizumab | 2 | 0.6 |
| Number of different DMARD | | |
| One medication | 329 | 97.1 |
| Two medications | 10 | 2.9 |

[a] Rheumatic disease types are listed in S2 Table

[b] DMARD types are listed in S1 Table

### Negative to positive IGRA conversion

Four patients were identified to have a conversion from negative to positive on repeated yearly IGRA testing (S4 Table). Patient A's positive TB1 mitogen value was 0.64 and TB2 was <0.10, repeat IGRA testing 43 days later showed TB1 mitogen value of 0.21 and TB2 0.16. Patient B's positive TB1 mitogen value was 0.54 and TB2 was 0.51, repeat IGRA testing 20 days later showed TB1 0.17 and TB2 0.14. As repeat IGRA testing yielded negative results in two patients (Patients A and B) and were adjudicated as a negative LTBI screen. Patient C had a prior history of LTBI treated with isoniazid (INH) for nine months and adjudicated as not having new LTBI diagnosis. Patient D was on secukinumab for psoriatic arthritis and was diagnosed with LTBI based on both positive IGRA test and known TB risk factors, including history of tobacco use and employment in healthcare setting. Patient D was treated for four months with rifampin. The calculated incidence of newly diagnosed LTBI over 5-years was 0.59 cases/ 1000 patient-years.

## Discussion

To the best of our knowledge, this is the first study to assess the utility of yearly IGRA screening among patients with rheumatic diseases on DMARD in the US, a non-TB endemic setting. In our 5-year experience, the incident of LTBI was rare in this population with negative baseline IGRA.

While it is recommended that all patients starting DMARD undergo screening for LTBI before initiating therapy [4], there is a lack of guidance on longitudinal screening of LTBI while on DMARD, particularly in non-TB endemic settings. Two studies from Greece, a non-TB endemic country per the WHO classification, demonstrated that about a third of patients with rheumatic diseases treated with TNF-alpha inhibitors or other biologics had initial negative LTBI screening tests that later converted to a positive screen [7, 8]. In one of these studies [8], 5 of 10 patients with a positive IGRA conversion received LTBI treatment. Reasons for non-treatment in the other half were not reported. Hence, it is unclear if treatment was deferred due to a medical contraindication, refusal, or not medically indicated (e.g., reversion to negative screen). A retrospective study from South Korea, a TB-endemic country, of patients with rheumatoid arthritis or ankylosing spondylitis on biologics found that 11.8% (N = 14/119) of patients had a conversion from negative to positive IGRA [9]. However, treatment decisions and outcomes were not described.

While we aimed to evaluate yearly IGRA tests, in practice few patients underwent yearly IGRA testing, suggesting logistic challenges in implementation of this approach. Furthermore, yearly screening for LTBI is faced with several challenges due to test performance variability. IGRAs can be falsely negative or indeterminate in immunocompromised hosts due to their blunted interferon-gamma responses [10]. Furthermore, serial IGRA testing of low-risk patients on DMARD may lead to a high rate of false-positive results in non-TB endemic areas [11]. Even immunocompetent hosts can have frequent IGRA negative to positive conversions and positive to negative reversions without new TB risk factors [12].

Hence, the current practice of universal, routine yearly IGRA-based screening may not be the effective approach to screen patients for LTBI in non-endemic settings. A suggested approach in non-TB endemic settings is targeted rather than universal screening of patients on DMARDs (S2 Fig). A targeted approach entails IGRA screening of those with vetted TB epidemiologic risk factors who should undergo IGRA testing [11].

We report several limitations to our study. First, this is a retrospective study and is limited by data documented in electronic medical records. Second, we only included patients with rheumatological diseases on DMARD; thus, the results are not generalizable to other

immunocompromised populations. Thirdly, our study was performed in a low TB-endemic area, so our findings may not apply to areas of high endemicity. Next, many patients did not have yearly IGRAs during the study time frame. Furthermore, we do not have data on all 340 patients' epidemiologic risk factors for TB. Lastly, since we used the problem list in the EMR recorded by physicians or other healthcare professionals to identify TB risk factors, we may have missed other risk factors documented in other sections of the EMR.

Yearly IGRA screening for LTBI in a non-endemic setting for TB was of low yield and only found one new LTBI case. Hence, a targeted screening approach based on TB epidemiologic risk factors prior to IGRA testing maybe a better approach. Future, prospective, multi-center studies are needed are needed to validate our findings and evaluate the cost-effectiveness of targeted screening of LTBI in this population living in TB non-endemic settings.

## Supporting information

**S1 Table. Disease Modifying Anti-Rheumatic Drugs (DMARD) categories.** Disease-modifying antirheumatic drugs (DMARDs) were grouped by mechanism of action.
(DOCX)

**S2 Table. Rheumatic disease categories.**
(DOCX)

**S3 Table. TB history.**
(DOCX)

**S4 Table. Characteristics of patients with conversion to positive QTF.**
(DOCX)

**S1 Fig. QuantiFERON-TB Gold testing results by year.**
(DOCX)

**S2 Fig. Suggested LTBI screening approach by TB epidemiologic risk factors.**
(DOCX)

## Author Contributions

**Conceptualization:** Vaidehi Chowdhary, Abhijeet Danve, Maricar Malinis.

**Data curation:** Carlo Foppiano Palacios, Ritche Hao.

**Formal analysis:** Carlo Foppiano Palacios.

**Methodology:** Carlo Foppiano Palacios, Maricar Malinis.

**Supervision:** Maricar Malinis.

**Writing – original draft:** Carlo Foppiano Palacios.

**Writing – review & editing:** Vaidehi Chowdhary, Ritche Hao, Abhijeet Danve, Maricar Malinis.

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
