## [Decision Letter · Decision Letter 0]

20 Dec 2023

PONE-D-23-30107Is yearly interferon gamma release assay latent tuberculosis infection screening warranted among patients with rheumatological diseases on disease-modifying drugs in non-endemic settings?PLOS ONE

Dear Dr. Malinis,

Thank you for submitting your manuscript to PLOS ONE. After careful consideration, we feel that it has merit but does not fully meet PLOS ONE’s publication criteria as it currently stands. Therefore, we invite you to submit a revised version of the manuscript that addresses the points raised during the review process.

We look forward to receiving your revised manuscript.

Kind regards,

Mao-Shui Wang

Academic Editor

PLOS ONE

Reviewers' comments:

Reviewer's Responses to Questions

**Comments to the Author**

1. Is the manuscript technically sound, and do the data support the conclusions?

Reviewer #1: Yes

Reviewer #2: Partly

Reviewer #3: No

2. Has the statistical analysis been performed appropriately and rigorously? 

Reviewer #1: Yes

Reviewer #2: N/A

Reviewer #3: No

3. Have the authors made all data underlying the findings in their manuscript fully available?

Reviewer #1: Yes

Reviewer #2: Yes

Reviewer #3: No

4. Is the manuscript presented in an intelligible fashion and written in standard English?

Reviewer #1: Yes

Reviewer #2: Yes

Reviewer #3: Yes

5. Review Comments to the Author

Reviewer #1: The authors present the results of a retrospective study assessing the value of annual retesting for TB in rheumatic patients in a low-endemicity country.

Some points for the authors to consider:

1. The authors report that patients were classified as: “having traveled to TB endemic countries” (page 5, ln 96-97). How was this information gathered?

2. The authors claim in the Discussion that “this is the first study to assess the utility of yearly IGRA screening among patients with rheumatic diseases on DMARD in the US, a non-TB endemic setting”. It should be clarified though, because it is not evident from the text, if all 615 patients had yearly IGRA testing. Is that the case? In page 7, ln 127-8 it is reported that “Only 18 patients had sequential yearly IGRA during the study period”. Please clarify.

3. In suppl table 4, 2/4 patients with a positive QFT converted back to negative. What was the interval period between the 2 measurements? This finding should be discussed further since it has been shown that in many cases this is a transient phenomenon in health care workers (Dorman SE et al, Am J Respir Crit Care Med 2014;189(1):77-87, Moses MW et al, Sci Rep 2016;6:30781.) as well as rheumatic patients. Do we have data regarding the value of QFT positivity in these patients? It is known that in many cases values close to the cut-off value of 0.35 can convert back to negative.

Reviewer #2: This is an interesting study demonstrating that the rigorous application of Guidelines regarding the screening for tuberculosis infection (TBI) in patients receiving biological therapies for rheumatologic diseases may conduct to unnecessary screening in a large majority of patient who have actually no risk of TBI and that selective screening based on individual risk factors for TBI may be more cost-effective. In fact, this is already the policy followed in some low TB incidence countries.

Comments:

1. The main comment regards the definition of risk factors for TBI. TB is a transmissible disease and nobody can be infected without having been in contact with a source case of TB. Having diabetes, using illegal drugs or being a smoker does not per se represent an increased risk for TBI or TB as long as the individual is not exposed to TB. On the other side, having a history of contact with TB or living or working in an environment with an increased risk of being exposed to TB is a risk factor for TBI, even in perfectly healthy individuals, but even more in diabetics. Therefore, I would suggest to the authors to re-analyse their data taking into account a) the history of patients with rheumatologic disease (prior contact with TB or working in an environment with increased risk), b) their origin (high TB incidence region), and evaluate if the selection of screening only individuals with such a risk factor is warranted. To note, it seems that the majority of patients included in this study were US-born patients with very little prior risk of TB contact and TBI, whereas in other regions of the world a large proportion of patients may be immigrants from high TB incidence regions, with a corresponding increase in the risk of TBI at baseline.

2. The second point which needs clarification is the policy regarding the 18 patients with a positive test result at initial screening, before the implementation of biological treatment. The authors should indicate if these patients were eligible for a preventive treatment and what are the criteria for it and not only for the 4 patients who converted from negative to positive. Considering the severity of TB in patients under biological therapies (mostly extensive forms), caution seems to be justified.

3. Most of the patients included in this study received a therapy which increases only marginally the risk of TB, except those 15% receiving TNF-alpha inhibitors. It would be interesting to analyze separately the patients under anti-TNF and the others.

Reviewer #3: Is yearly interferon gamma release assay latent tuberculosis infection screening warranted among patients with rheumatological diseases on disease-modifying drugs in non-endemic settings?

First of all I would like to thank you for the opportunity to review this interesting paper on LTBI/IGRA

About the article there are some points that I would like to highlight for its possible publication;

In material and methods it is described that an annual IGRA will be performed, but finally only 18 patients had sequential yearly IGRAs, it would be important to specify how many patients underwent the IGRA at least every two years, or how often it was performed. .

There is no table in the results that explains this.

If sequential follow-up was only performed on 18 patients, it is difficult to extrapolate that only four converted on a yearly basis.

Given the risk of developing active TB in these patients, it would also be necessary to explain in more detail why only one was treated.

The number of patients with anti-TNF drugs in the total population is low

Conclusion:

Retrospective study that is based on the description of the number of IGRA conversions in patients with Rheumatological disease. The main approach, which is to carry out an annual IGRA, is only fulfilled in 18/615; it would be necessary to expand this information with the average number of years in which the IGRAs were carried out consecutively.

6. PLOS authors have the option to publish the peer review history of their article (what does this mean?). If published, this will include your full peer review and any attached files.

Reviewer #1: No

Reviewer #2: **Yes: **Jean-Pierre Zellweger, MD, Switzerland

Reviewer #3: No

---

## [Author Response · Author response to Decision Letter 0]

18 Mar 2024

February 23, 2024 

To the Editor and Reviewer

Thank you for the opportunity to revise our submitted manuscript – “Is repeat interferon gamma release assay latent tuberculosis infection screening warranted among patients with rheumatological diseases on disease-modifying drugs in non-endemic settings?” Kindly find below our point-by-point responses to editors’ and reviewers’ comments. 

Thank you for your consideration!

REVIEWER COMMENTS:

Reviewer #1: The authors present the results of a retrospective study assessing the value of annual retesting for TB in rheumatic patients in a low-endemicity country.

Some points for the authors to consider:

1. The authors report that patients were classified as: “having traveled to TB endemic countries” (page 5, ln 96-97). How was this information gathered?

Response: Thank you for the comments. Travel to endemic country was gathered from review of documentation within the electronic medical record.

2. The authors claim in the Discussion that “this is the first study to assess the utility of yearly IGRA screening among patients with rheumatic diseases on DMARD in the US, a non-TB endemic setting”. It should be clarified though, because it is not evident from the text, if all 615 patients had yearly IGRA testing. Is that the case? In page 7, ln 127-8 it is reported that “Only 18 patients had sequential yearly IGRA during the study period”. Please clarify.

Response: Thank you for the comments and suggestions. We have limited the inclusion criteria by only including patients with at least two IGRAs during the study period (N=340). We have added the following to the results section, “While patient with rheumatic diseases on DMARDs were supposed to undergo yearly IGRA testing, data demonstrated 340 patients underwent IGRA testing at least twice during the study period including 51.8% (N=176) with at least 3 IGRA tests, 18.5% (N=63) with at least 4 IGRA tests, and 5.3% (N=18) IGRA testing every year during the study period.”

3. In suppl table 4, 2/4 patients with a positive QFT converted back to negative. What was the interval period between the 2 measurements? This finding should be discussed further since it has been shown that in many cases this is a transient phenomenon in health care workers (Dorman SE et al, Am J Respir Crit Care Med 2014;189(1):77-87, Moses MW et al, Sci Rep 2016;6:30781.) as well as rheumatic patients. Do we have data regarding the value of QFT positivity in these patients? It is known that in many cases values close to the cut-off value of 0.35 can convert back to negative.

Response: Thank you for the comments and suggestions. We have added the following, “Patient A’s positive TB1 mitogen value was 0.64 and TB2 was <0.10, repeat IGRA testing 43 days later showed TB1 mitogen value of 0.21 and TB2 0.16. Patient B’s positive TB1 mitogen value was 0.54 and TB2 was 0.51, repeat IGRA testing 20 days later showed TB1 0.17 and TB2 0.14. As repeat IGRA testing yielded negative results in two patients (Patients A and B) and were adjudicated as a negative LTBI screen.”

Reviewer #2: This is an interesting study demonstrating that the rigorous application of Guidelines regarding the screening for tuberculosis infection (TBI) in patients receiving biological therapies for rheumatologic diseases may conduct to unnecessary screening in a large majority of patient who have actually no risk of TBI and that selective screening based on individual risk factors for TBI may be more cost-effective. In fact, this is already the policy followed in some low TB incidence countries.

Comments:

1. The main comment regards the definition of risk factors for TBI. TB is a transmissible disease, and nobody can be infected without having been in contact with a source case of TB. Having diabetes, using illegal drugs or being a smoker does not per se represent an increased risk for TBI or TB as long as the individual is not exposed to TB. On the other side, having a history of contact with TB or living or working in an environment with an increased risk of being exposed to TB is a risk factor for TBI, even in perfectly healthy individuals, but even more in diabetics. Therefore, I would suggest to the authors to re-analyse their data taking into account a) the history of patients with rheumatologic disease (prior contact with TB or working in an environment with increased risk), b) their origin (high TB incidence region), and evaluate if the selection of screening only individuals with such a risk factor is warranted. To note, it seems that the majority of patients included in this study were US-born patients with very little prior risk of TB contact and TBI, whereas in other regions of the world a large proportion of patients may be immigrants from high TB incidence regions, with a corresponding increase in the risk of TBI at baseline.

Response: Thank you for the comments and suggestions. We agree with the reviewer that certain co-morbidities put patients at risk for TB-reactivation, but epidemiologic risk factors are the most relevant in terms of exposure to TB. TB screening of epidemiologic risk factors is as important as the TST or IGRA. Unfortunately, we do not have the data on all (N=340) patients’ epidemiologic TB risk factors for the suggested analysis, we have added this as a limitation, “Furthermore, we do not have data on all 340 patients’ epidemiologic risk factors for TB.” However, using primary language other than English as a marker for born outside of the U.S., only 13 patients (3.8%) spoke a primary langue other than English. We have changed the wording on Table S3 to reflect “TB predisposing factors” rather than epidemiologic TB risk factors.” Additionally, we have added the TB epidemiologic TB risk factors for the 18 patients with yearly IGRA, “Among the 18 patients with yearly IGRA, 4 (22.2%) had new travel to a TB-endemic setting.” 

2. The second point which needs clarification is the policy regarding the 18 patients with a positive test result at initial screening, before the implementation of biological treatment. The authors should indicate if these patients were eligible for a preventive treatment and what are the criteria for it and not only for the 4 patients who converted from negative to positive. Considering the severity of TB in patients under biological therapies (mostly extensive forms), caution seems to be justified.

Response: Thank you for the comments and suggestions. We agree with your comment that epidemiologic exposure is important data to determine who may benefit from LTBI therapy. In our study cohort, only 18 patients had yearly IGRA testing completed. Separately, there were only 4 patients who had a conversion from a negative to a positive IGRA test during our study time frame: two with repeat negative IGRA and no TB risk factors, one with prior treated LTBI, and one with TB risk factors and new LTBI.

3. Most of the patients included in this study received a therapy which increases only marginally the risk of TB, except those 15% receiving TNF-alpha inhibitors. It would be interesting to analyze separately the patients under anti-TNF and the others.

Response: Thank you for the comments and suggestions. In addition to TNF-alpha inhibitors, we would also include JAK inhibitors as high-risk targeted therapies for TB. Among patients on TNF-alpha inhibitors & JAK inhibitors (N=129/340, 37.9%), there was only one seroconversion from negative to positive IGRA in a patient with prior LTBI.

Reviewer #3: Is yearly interferon gamma release assay latent tuberculosis infection screening warranted among patients with rheumatological diseases on disease-modifying drugs in non-endemic settings?

First of all, I would like to thank you for the opportunity to review this interesting paper on LTBI/IGRA

About the article there are some points that I would like to highlight for its possible publication.

In material and methods it is described that an annual IGRA will be performed, but finally only 18 patients had sequential yearly IGRAs, it would be important to specify how many patients underwent the IGRA at least every two years, or how often it was performed. 

There is no table in the results that explains this.

If sequential follow-up was only performed on 18 patients, it is difficult to extrapolate that only four converted on a yearly basis.

Response: Thank you for the comments and suggestions. We have limited the inclusion criteria by only including patients with at least two IGRAs during the study period (N=340). Furthermore, 51.8% (N=176/340) had at least 3 IGRA tests, 18.5% (N=63/340) had at least 4 IGRA tests, and 5.3% (N=18/340) IGRA testing every year during the study time frame.

Given the risk of developing active TB in these patients, it would also be necessary to explain in more detail why only one was treated.

Response: Thank you for the comments and suggestions. There were only 4 patients who had a conversion from a negative to a positive IGRA test during our study time frame: two who had repeat IGRA that was negative after the positive and no epidemiologic TB risk factors, one with known prior LTBI treatment, and one with TB risk factors and new LTBI.

The number of patients with anti-TNF drugs in the total population is low

Response: Thank you for the comment. In addition to TNF-alpha inhibitors, we would also include JAK inhibitors as high-risk targeted therapies for TB. Overall, 37.9% of patients (N=129/340) were on TNF-alpha inhibitors & JAK inhibitors.

Conclusion:

Retrospective study that is based on the description of the number of IGRA conversions in patients with Rheumatological disease. The main approach, which is to carry out an annual IGRA, is only fulfilled in 18/615; it would be necessary to expand this information with the average number of years in which the IGRAs were carried out consecutively.

Response: Thank you for the comments and suggestions. We have addressed this issue in the prior response.

---

## [Decision Letter · Decision Letter 1]

3 Apr 2024

PONE-D-23-30107R1Is yearly interferon gamma release assay latent tuberculosis infection screening warranted among patients with rheumatological diseases on disease-modifying drugs in non-endemic settings?PLOS ONE

Dear Dr. Malinis,

Thank you for submitting your manuscript to PLOS ONE. After careful consideration, we feel that it has merit but does not fully meet PLOS ONE’s publication criteria as it currently stands. Therefore, we invite you to submit a revised version of the manuscript that addresses the points raised during the review process.

We look forward to receiving your revised manuscript.

Kind regards,

Mao-Shui Wang

Academic Editor

PLOS ONE

Journal Requirements:

Reviewers' comments:

Reviewer's Responses to Questions

**Comments to the Author**

1. If the authors have adequately addressed your comments raised in a previous round of review and you feel that this manuscript is now acceptable for publication, you may indicate that here to bypass the “Comments to the Author” section, enter your conflict of interest statement in the “Confidential to Editor” section, and submit your "Accept" recommendation.

Reviewer #1: All comments have been addressed

Reviewer #2: All comments have been addressed

2. Is the manuscript technically sound, and do the data support the conclusions?

Reviewer #1: Yes

Reviewer #2: Yes

3. Has the statistical analysis been performed appropriately and rigorously? 

Reviewer #1: Yes

Reviewer #2: I Don't Know

4. Have the authors made all data underlying the findings in their manuscript fully available?

Reviewer #1: Yes

Reviewer #2: Yes

5. Is the manuscript presented in an intelligible fashion and written in standard English?

Reviewer #1: Yes

Reviewer #2: Yes

6. Review Comments to the Author

Reviewer #1: (No Response)

Reviewer #2: The authors have addressed all comments except one:

Please specify if the 18 patients with a positive IGRA at first screening were eligible for preventive therapy and, if not, why

Comment from review #2: The second point which needs clarification is the policy regarding the 18 patients with a

positive test result at initial screening, before the implementation of biological treatment. The

authors should indicate if these patients were eligible for a preventive treatment and what are

the criteria for it

7. PLOS authors have the option to publish the peer review history of their article (what does this mean?). If published, this will include your full peer review and any attached files.

Reviewer #1: No

Reviewer #2: No

---

## [Author Response · Author response to Decision Letter 1]

9 Jun 2024

To the Editor and Reviewers

We are truly grateful for the review and comments of the reviewers. We revised the paper to clarify the comments of the reviewer, which we think also improved the paper’s clarity in describing the results.

Below is our response to the reviewer’s comment ( in blue) : 

Review Comments to the Author

Reviewer #2: The authors have addressed all comments except one:

Reviewer #2: Could you please specify if the 18 patients with a positive IGRA at first screening were eligible for preventive therapy and, if not, why?

We want to address a crucial point raised by Reviewer 2. Of the 339 patients who tested at least twice during the study period, only 18 had documented yearly testing as described in the demographics section ( line 101, page 6). In 2017 (the start of the study), only 2 tested positive for IGRA, and both had prior documented LTBI treatment (page 7, line 115-116). The abstract was also revised to make it clear of results at baseline. 

We revised the tuberculosis history and IGRA results to clarify the circumstances of the positive testing. We created a subsection of those who converted from negative to positive tests and the course thereafter. 

All revisions were highlighted in yellow. Thank you again for the comment. 

 Comment from review #2: The second point which needs clarification is the policy regarding the 18 patients with a positive test result at initial screening, before the implementation of biological treatment. The authors should indicate if these patients were eligible for a preventive treatment and what are the criteria for it.

Please see the response to the prior comment. 

Sincerely,

Maricar Malinis

Carlos Fopianno Palacios

---

## [Editor Report · Decision Letter 2]

13 Jun 2024

Is yearly interferon gamma release assay latent tuberculosis infection screening warranted among patients with rheumatological diseases on disease-modifying drugs in non-endemic settings?

PONE-D-23-30107R2

Dear Dr. Malinis,

We’re pleased to inform you that your manuscript has been judged scientifically suitable for publication and will be formally accepted for publication once it meets all outstanding technical requirements.

Kind regards,

Mao-Shui Wang

Academic Editor

PLOS ONE
---

## [Editor Report · Acceptance letter]

24 Jun 2024

PONE-D-23-30107R2 

PLOS ONE

Dear Dr. Malinis, 

I'm pleased to inform you that your manuscript has been deemed suitable for publication in PLOS ONE. Congratulations! Your manuscript is now being handed over to our production team.

Kind regards, 

on behalf of

Dr. Mao-Shui Wang 

Academic Editor

PLOS ONE